



# Enhancing Parameter Calibration in Land Surface Models Using a Multi-Task Surrogate Model within a Differentiable Parameter Learning Framework

Wenpeng Xie[1], Hongmei Li[2], Kei Yoshimura[1,3]

[1]Institute of Industrial Sciences, The University of Tokyo, Kashiwa, Japan.
[2]Department of Natural Environmental Studies, Graduate School of Frontier Sciences, The University of Tokyo, Kashiwa, Japan.
[3]Earth Observation Research Center, Japan Aerospace Exploration Agency, Tsukuba, Japan.

*Correspondence to*: Wenpeng Xie (wenpeng@iis.u-tokyo.ac.jp)

**Abstract.** Land surface models (LSMs) are essential for simulating terrestrial processes and their interactions with the atmosphere. However, parameter calibration in LSMs remains a major challenge owing to complex process coupling and parameter uncertainty. For example, key parameters, such as plant function type (PFT), are often estimated using field measurements or empirical relationships, which are characterized by limited accuracy, resulting in systematic biases and inconsistencies. In this study, we introduce multiple-task differentiable parameter learning (MdPL), a deep learning framework that combines a multitask surrogate model with a differentiable parameter generator for more accurate and efficient LSM parameter calibration. The multitask surrogate learns both shared and task-specific features to predict multiple fluxes, and the differentiable generator infers site-specific parameters from meteorological forcings and land surface attributes. Calibrated across 20 sites spanning four PFTs, the MdPL-calibrated Integrated Land Simulator (ILS) achieved a 15% decrease in RMSE for both sensible and latent heat flux simulations. Further, benchmarking using the PLUMBER2 dataset showed that the MdPL-calibrated ILS outperformed standard LSMs (CLM5, JULES, Noah, and GFDL), and its accuracy matched or exceeded those of LSTM-based approaches. The assessment of its transferability via leave-one-out cross-validation for evergreen forest, woodland, and cultivation sites showed reasonable applicability across sites for evergreen forests and woodlands, with parameter sets yielding close-to-optimal flux simulations, even without site specification. However, for cultivation sites, PFT parameters exhibited strong site specificity, with parameter sets from the same PFT not reliably transferred. Despite its reduced effectiveness of the framework for cultivation sites under fixed PFT settings, it offers a scalable and physically grounded approach for enhancing parameter calibration in complex LSMs.

## 1 Introduction

To address the unprecedented challenges posed by climate change, extreme weather, and biodiversity loss, it is critical to accurately forecast terrestrial ecosystem dynamics under these global change scenarios (Cardinale et al., 2012; Raoult et al., 2024; Rivera et al., 2017). Land surface models (LSMs) are key components of Earth System Models that provide





mechanistic simulations of land-atmosphere interactions (Blyth et al., 2021). They also capture biogeochemical cycles, water fluxes, and energy exchanges. Thus, offer valuable insights into land management, water management, and agricultural

production at the global scale for ecosystems (Nitta et al., 2020). However, ecosystem complexity, driven by spatial heterogeneity and nonlinear process couplings, introduce significant uncertainties in LSM parameterization (Fisher & Koven, 2020; Li et al., 2024).

Further, LSM parameters often lack direct observational constraints (Famiglietti et al., 2021), exhibit multiscale dependencies, and interact nonlinearly within coupled systems. For example, soil hydraulic conductivity and minimum

stomatal conductance are often estimated empirically or using sparse field measurements, which introduce biases and inconsistencies (Buotte et al., 2021; Exbrayat et al., 2014; Oberpriller et al., 2022). Moreover, simplifying model processes, such as radiation transfer or turbulent exchange schemes for wind speed, can introduce systematic biases, which can further complicate parameter calibration (Sawada, 2020).

For a long time, manual parameter tuning was the only strategy for calibrating LSM parameters, and owing to the limited

availability of computational resources and observational data, module parameters were primarily derived from existing knowledge (Blyth et al., 2021). However, over the past two decades, advances in computing power have enhanced parameter data assimilation (PDA), which combines observational data with model states to update parameters iteratively (Medvigy et al., 2009; Rayner et al., 2005). PDA also synergistically fuses observations and model forecasts within a probabilistic framework and thus, simultaneously quantifies data and models formulation uncertainties (Rayner et al., 2019). This

approach has yielded substantial progress in the calibration of parameters related to crop, carbon, and hydrological cycles (Kaminski et al., 2002; Keenan et al., 2013; Peylin et al., 2016; Weng et al., 2011). However, it is associated with several limitations, particularly in the context of LSMs. Additionally, its high computational cost increases exponentially with the number of parameters. Thus, it is impractical for complex systems (Bacour et al., 2023; Raiho et al., 2021; Schürmann et al., 2016). Its stability and accuracy in converging optimal parameter sets are also affected by nonlinear processes, such as soil

moisture feedback and energy exchange (Huang et al., 2018; Massoud et al., 2019). Moreover, its effectiveness is limited by its dependence on dense and high-quality observations, meanwhile existing observations are often sparse and noisy across space and time (Cameron et al., 2022; MacBean et al., 2016).

Machine learning (ML) offers novel possibilities for addressing PDA-associated limitations. Notably, ML methods can be employed to accurately capture complex and nonlinear relationships and optimize high-dimensional parameter spaces,

thereby offering new avenues for augmenting PDA workflow (Chen et al., 2022; Kolassa et al., 2017; Kwon et al., 2019; Rodríguez-Fernández et al., 2019). To date however, most ML efforts have served as adjuncts within PDA frameworks, rather than being the primary learning objective responsible for parameter estimation (McNeall et al., 2024). Tsai et al. (2021) introduced differentiable parameter learning (dPL), an end-to-end deep learning scheme that directly embeds parameter estimation within model training to enhance calibration efficiency. For efficient parameter inference, dPL, e.g., the

variable infiltration capacity (VIC) model, leverages neural-network differentiability to couple process-based simulators (Liang et al., 1994) with gradient-based training. In this end-to-end workflow, the neural network, $g\_z$ (·) uses forcings,



historical site data (forcings + flux observations), and static land surface properties to estimate the parameters of the VIC model. The predicted parameters were used to drive a neural surrogate of the VIC and the resulting outputs were then compared with observation data, enabling backpropagation-based calibration. Thus, dPL is highly advantageous across
different datasets, such as soil moisture and runoff. Similarly, Feng et al. (2023) applied dPL to calibrate the parameters of the Hydrologiska Byråns Vattenbalansavdelning hydrological model (Tibangayuka et al., 2022). Their results showed notable improvements in the accuracy of soil and groundwater storage, snow accumulation, evapotranspiration, and baseflow simulations across different basins in the United States. Therefore, relative to PDA models, dPL is advantageous in that it offers the possibility to transform the parameter calibration problem into a multi-model problem and uses deep learning
frameworks to optimize parameters efficiently via backpropagation and gradient descent algorithms (Shen et al., 2023). Consequently, the parameters for every site are learned in one pass, minimizing manual tuning and improving model transferability. Second, by exploiting model differentiability, dPL sidesteps costly repeated simulator runs and reduces calibration time by several orders of magnitude relative to the cost and time requirements associated with PDA (Tsai et al., 2021). dPL's gradient-based approach also adeptly learns nonlinear couplings, which are critical for processes such as snow-
soil interactions in hydrology (Feng et al., 2022).

However, applying dPL in LSMs is associated with several limitations. First, surrogate fidelity sets the ceiling on parameter accuracy, and any mismatch between the neural surrogate and the true LSM increases calibration errors. Second, LSMs, unlike standalone hydrological simulators, integrate energy fluxes, carbon cycling, and vegetation physiology, resulting in a significant increase in emulation difficulty (MATSIRO6 Document Writing Team, 2021). This multidomain coupling and
strong nonlinearity complicate surrogate architecture and training. Third, simultaneously calibrating multiple fluxes and states in LSMs requires the careful mapping of parameters to each target variable. Further, different parameters exhibit varying levels of sensitivity and impact on each output. The effects of coupling and interactions on different processes also amplify uncertainties in parameter calibration (Hou et al., 2015). Therefore, the key challenge associated with applying dPL in LSMs is the design of surrogate models and management of model complexity.

To address the multi-process and multi-output complexity of LSMs with respect to dPL, in this study, we propose a multiple-task differentiable parameter learning (MdPL) framework as a unified calibration framework for application in LSMs. The MdPL framework comprised two modules, namely, a multi-task surrogate and a differentiable parameter generator, $g\_z$ (·). Based on multitask learning (Ruder, 2017), the multi-task surrogate uses shared bases to capture common dynamics, and thereafter employ separate task-specific branches to fine-tune the parameters of each target flux. We also integrated gate
layers that filter and route parameter signals and automatically emphasize those relevant to each output, while suppressing irrelevant ones, thereby reducing overfitting. Further, the pipeline parallel, $g\_z$ (·) was employed to generate parameters that drive the multi-task surrogate, and its predictions were compared against observed fluxes for end-to-end optimization. Before calibration, we pretrained the surrogate using data synthesized by perturbing the key parameters, ensuring that it learned their influence on flux outputs.





Compared with single-task dPL, our multitask design offered the possibility to reduce cross-output interference and handle integrated process complexity, providing a more efficient and robust approach for calibrating parameters in complex land surface models. Further, to validate the performance of the proposed framework, we used it to calibrate the Integrated Land Simulator (ILS) across 20 PLUMBER2 sites, representing four plant functional types (evergreen forest, woodland, cropland, and grassland) and thereafter, benchmarked the MdPL-calibrated ILS against four leading LSMs, CLM5, JULES, Noah, and

GFDL, and a Long Short-Term Memory (LSTM) model using half-hourly sensible and latent heat observations. We also evaluated the transferability of the calibrated ILS across different PFTs via leave-one-out cross-validation (LOOCV).

Thus, we present MdPL as an efficient calibration framework that addresses process complexity, multi-output coupling, and parameter uncertainty, which limit LSM accuracy. Further, using the multitask learning-inspired surrogate model, we accurately captured nonlinear and coupled processes, reduced computational costs through gradient-based optimization, and

mitigated the impact of sparse and noisy observational data. Together, these improvements boost the generalizability and reliability of LSMs, providing a scalable calibration strategy for ecosystem simulations under climate change. The rest of the manuscript is structured as follows. Section 2 provides details regarding the MdPL framework, evaluation metrics, data preprocessing, and the experiments performed. Sect. 3 provides results and discussion, and Sect. 4 provides a conclusion of the study, summarizing the key findings of the study and their implications.



## 2 Materials and Methods

### 2.1 Framework outline

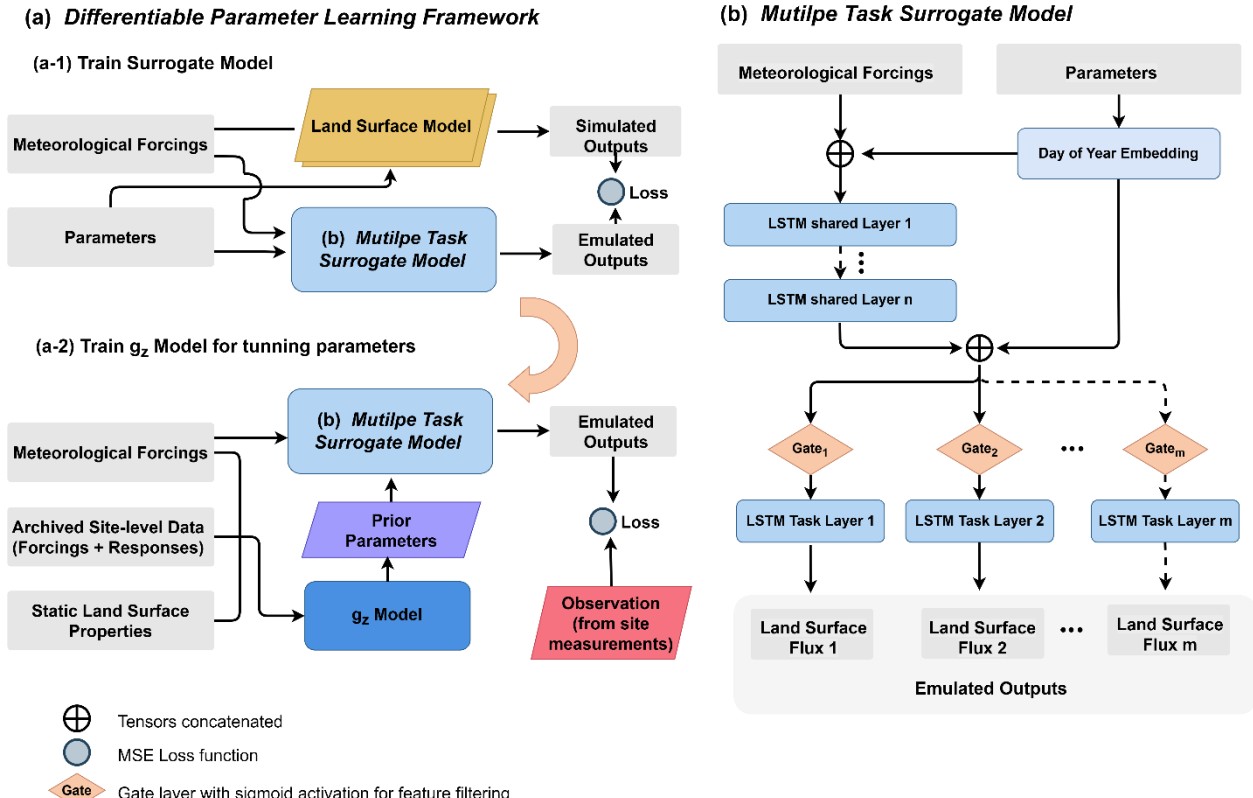

**Figure 1. Overview of the differentiable parameter learning (dPL) framework for land surface models (LSMs). The surrogate model emulated LSM outputs based on meteorological forcings and parameters. A secondary model, (gZ) inferred site-specific parameter priors using static land surface properties and historical site data (forcings + flux observations). Loss was computed by comparing surrogate predictions and observed fluxes.**

The MdPL framework is illustrated in **Fig. 1(a)**, from which it is evident that the calibration of LSM parameters was performed in two main steps, the first being to train the multiple-task surrogate model for the LSM, as shown in **Fig. 1(a1)**, ensuring that the surrogate model replicated the LSM outputs as closely as possible.

$$\min_{\theta} \frac{1}{N} \sum_{i=1}^{N} \mathcal{L}(\hat{f}(X_i, P_i, Encoding(DOY); \theta), f(X_i, P_i)) \tag{1}$$

In Eq. (1), $X_i$ represents the meteorological forcing of the $i$-th sample, $P_i$ represents the calibration parameters of the $i$-th sample, $Encoding(DOY)$ represents the time positional encoding of the day of the year, $f(X_i, P_i)$ denotes the LSM output, $\hat{f}(X_i, P_i; \theta)$ denotes the predicted outputs of the surrogate model, with $\theta$ referring to the trainable parameters (e.g., weights and biases) of the neural network. Further, $\mathcal{L}$ represents the loss function, typically expressed as the mean squared error (MSE), and $N$ represents the total number of training samples.





The surrogate model, which was trained using the same inputs as those in the LSM, i.e., meteorological forcing and calibration parameters, with LSM outputs as the target data, learned the LSM input-output mapping pattern by minimizing the error between predictions and targets. Further, surrogate models established based on differentiable functions, similar to neural networks, were employed to approximate LSM behavior and support automatic differentiation. Thus, they played critical roles in gradient-based optimization in MdPL.

The second step involved learning the optimal LSM parameters using a parameter generation model, $g_z$, as shown in **Fig. 1(a2)**. Specifically, $g_z$ is a deep learning framework with meteorological forcing data, $X$ and auxiliary information, $A$ (archived site-level data and land surface properties) as inputs and it provides calibration parameters as outputs.

$$\min_{\emptyset} \frac{1}{N} \sum_{j=1}^{N} \mathcal{L}(\hat{f}(X_j, g_z(X_j, A_j; \emptyset), Encoding(DOY); \bar{\theta}), Y_{obs,j}) \tag{2}$$

In Eq. (2), $A_j$ represents archived site-level data and land surface properties for the $j$-th sample, $g_z(X_j, A_j; \emptyset)$ denotes calibration parameters generated by the parameter generation model, $\emptyset$ represents the trainable parameters in the parameter generation model, $\bar{\theta}$ represents the frozen parameters of the surrogate model, and $Y_{obs,j}$ represents the observed data for the $j$-th sample. This learning process leverages the parameter generation model, $g_z$ to map the observational data and dynamic forcing inputs to the LSM parameter space, thereby enabling automated parameter calibration.

## 2.2 Multiple-task Surrogate Model

Details regarding the multiple-task surrogate model are shown in **Fig. 1(b)**. Unlike dPL, which directly concatenates forcing data and calibration parameters as inputs, the multiple-task surrogate model adopts a different processing strategy. Direct concatenation may hinder the ability of the model to effectively differentiate between these two types of features, increase feature learning complexity, and reduce key process simulation accuracy (Leontjeva & Kuzovkin, 2016). Further, differences in scale and variation frequency between static and dynamic features can interfere with the extraction of critical parameter information, ultimately limiting the overall performance of the surrogate model (Han et al., 2022).

To capture the influence of temporal variations on LSM parameters, we introduced time positional encoding (TPE) based on the "day of year" (DOY) (Foumani et al., 2024). This encoding method transformed temporal information into high-dimensional feature representations, and thus, enhanced the ability of the model to perceive seasonal variability. The TPE was computed according to Eqs. (3) and (4)

$$DOY_{rad} = DOY \cdot \frac{2\pi}{365} \tag{3}$$

$$Encoding(DOY) = [sin(f_l \cdot DOY_{rad}), cos(f_l \cdot DOY_{rad})]_{l=0}^{L-1} \tag{4}$$

where $f_l = 2^l$ represents the frequency of each encoding dimension, with $l \in \{0, 1, \dots, L-1\}$. TPE also captured multiscale temporal periodicity and provided a robust representation of seasonal effects on model parameters.

In this study, to extract global features from dynamic inputs, we employed a LSTM network as a shared layer (Hochreiter & Schmidhuber, 1997). The output features of the LSTM, denoted as $H_{shared}$, were expressed as shown in Eq. (5).

$$H_{shared} = LSTM_{shared}(X_{combined}) \tag{5}$$





The dynamic forcing data, $X_{\text{forcing}}$, time positional encoding, $Encoding(DOY)$, and calibration parameters, $P$ were concatenated to form the combined feature representation, denoted $X_{\text{combined}}$ and expressed according to Eq. (6).

$$X_{\text{combined}} = [X_{\text{forcing}}, \text{Encoding}(DOY), P] \tag{6}$$

Further, to address task competition in multitask learning and reduce the influence of parameters that are irrelevant to specific outputs, a gate layer was introduced before the task-specific layers. This gate layer dynamically adjusted the flow of features based on the input as defined in Eqs. (7), (8), and (9).

$$g_i = \sigma\left(W_{\text{gate},i} \cdot X_{\text{combined}} + b_{\text{gate},i}\right) \tag{7}$$

$$X_{\text{gated},i} = g_i \odot H_{\text{shared}} \tag{8}$$

$$Y_{\text{pre},i} = LSTM_{\text{Task},i}(X_{\text{gated},i}) \tag{9}$$

where $W_{\text{gate},i}$ and $b_{\text{gate},i}$ represent the weights and biases of the gate layer for the $i$-th task, respectively, $\sigma$ denotes the sigmoid activation function, and $\odot$ represents element-wise multiplication. The gated features, $X_{\text{gated}}$ were then fed into each task-specific layer to learn the features relevant to different output objectives. $Y_{\text{pre},i}$ represents the output of model task $LSTM_{\text{Task},i}$ for the $i$-th task.

The LSTM shared layer extracted global features from the time series and thus, provided a common representation for multitask learning. The task-specific layers, combined with the gate layer, effectively mitigated competition among tasks and enhanced the ability of the model to fit individual task objectives. Further, by dynamically controlling feature flow, the gate layer limited the impact of irrelevant information on the training process, thereby improving the stability and generalizability of the model.

## 2.3 Evaluation Metrics

To comprehensively assess the performance of the MdPL framework, we utilized a range of evaluation metrics to obtain insights regarding model accuracy, consistency, and predictive reliability. The metrics employed included the Root Mean Square Error (RMSE), which indicates the average difference between predicted and observed values, giving more weight to larger errors because of the squaring in its calculation. This metric was calculated according to Eq. (10) as shown below.

$$RMSE = \sqrt{\frac{1}{n}\sum_{i=1}^{n}(y_i - \hat{y}_i)^2} \tag{10}$$

where $y_i$ and $\hat{y}_i$ represent observed and predicted values, respectively, and $n$ represents number of samples. Lower RMSE values were indicative of better model performance. The second metric used was standard deviation (STD), which provides a measure of the variability of the predicted values. It was determined according to Eq. (11) as follows:

$$STD = \sqrt{\frac{1}{n}\sum_{i=1}^{n}(\hat{y}_i - \bar{\hat{y}}_i)^2} \tag{11}$$



where $\overline{\hat{y}_i}$ denotes the mean of the predicted values. A predicted STD closer to the observed STD implied that the model accurately captured the natural variability of the data. Further, correlation coefficient values, $R$, were calculated to capture the degree of linear correlation between the observed and predicted values. The values were calculated according to Eq. (12).

$$R = \frac{\sum_{i=1}^{n}(y_i - \bar{y}_i)(\hat{y}_i - \overline{\hat{y}_i})}{\sqrt{\sum_{i=1}^{n}(y_i - \bar{y}_i)^2}\sqrt{\sum_{i=1}^{n}(\hat{y}_i - \overline{\hat{y}_i})^2}} \tag{12}$$

$R$ values closer to 1 were indicative of a strong positive correlation, whereas values closer to -1 were indicative a strong negative correlation. Kling–Gupta efficiency (KGE), an advanced performance metric that combines correlation, bias, and variability into a single score (Gupta et al., 2009), was also employed in this study. It was determined according to Eq. (13).

$$KGE = 1 - \sqrt{(r-1)^2 + (\beta - 1)^2 + (\gamma - 1)^2} \tag{13}$$

where $r$ represents correlation coefficient, $\beta$ represents the ratio of the mean predicted and observed values, and $\gamma$ represents

the ratio of the predicted and observed standard deviations. KGE values close to 1 were indicative of better performance. Nash–Sutcliffe efficiency (NSE), which provides a measure of how well predicted values match observed data relative to the baseline model, was also employed in this study. It was determined according to Eq. (14).

$$NSE = 1 - \frac{\sum_{i=1}^{n}(y_i - \hat{y}_i)^2}{\sum_{i=1}^{n}(y_i - \bar{y}_i)^2} \tag{14}$$

NSE values close to 1 were indicative of a better fit, whereas negative values indicated that the mean of the observed data

performs better than that of the model.

## 2.4 Data and Experiments

### 2.4.1 Data sources

The PLUMBER2 dataset (PALS Land Surface Model Benchmarking Evaluation Project), a public framework for the standardized inter-comparison of LSMs and data-driven models, was used to simulate energy, water, and carbon fluxes

across global ecosystems (Abramowitz et al., 2024). The dataset comprises: (1) meteorological forcings, (2) site attributes, (3) flux observations, and (4) model outputs spanning LSMs and data-driven methods (e.g., linear regression, ML, Deep Learning). Thus, it provides a comprehensive foundation for benchmarking.

To ensure geographic and ecosystem diversity, we selected 20 globally distributed sites representing four PFTs: broadleaf evergreen forests (in this study, evergreen forests), mixed coniferous and broadleaf deciduous forests and woodlands

(woodland), wooded and grassland (grassland), and cultivation sites (Details regarding these PFTs are provided in Table 1. Available half-hourly flux measurements, averaging 3 years of continuous data for each site, were also used. Site attributes included soil texture, vegetation classification, and geographic coordinates. To unify the influences of various variables on the neural network, the forcing and outputs were standardized using means and STDs.

**Table 1. Summary of the characteristics of the observation sites for different plant function types.**



| Plant function type | Site name | Location (lon, lat) | Observation period | Climate | Country and region |
|---|---|---|---|---|---|
| Broadleaf evergreen forest | CN-Din | 23.17, 112.54 | 2003 - 2005 | Humid Subtropical | China |
| | ID-Pag | -2.32, 113.90 | 2002 - 2003 | Tropical rain forest | Indonesia |
| | PT-Esp | 38.64, -8.60 | 2002 - 2004 | SubTropical | Portugal |
| | PT-Mi1 | 38.54, -8.00 | 2005 - 2005 | SubTropical | Portugal |
| Mixed coniferous & broadleaf deciduous forest & woodland | AR-SLu | -33.46, -66.46 | 2010 -2010 | SubTropical | Argentina |
| | CN-Cha | 42.40, 128.10 | 2003 - 2005 | Temperate | China |
| | DE-Meh | 51.28, 10.66 | 2004 - 2006 | Temperate | Germany |
| | JP-SMF | 35.26, 137.08 | 2003 - 2006 | SubTropical | Japan |
| | US-Bar | 44.06, -71.29 | 2005 - 2005 | Temperate | USA |
| | AU-Emr | -23.86, 148.47 | 2012 - 2013 | | Australia |
| Wooded & grassland | CN-Dan | 30.85, 91.08 | 2004 - 2005 | Arctic | China |
| | CN-Du2 | 42.05, 116.28 | 2007 - 2008 | Temperate | China |
| | DK-Lva | 55.68, 12.08 | 2005 - 2006 | Temperate | Denmark |
| | IE-Dri | 51.99, -8.75 | 2003 - 2005 | Temperate | Ireland |
| | PL-wet | 52.76, 16.31 | 2004 - 2005 | Temperate | Poland |
| Cultivation land | DE-Seh | 50.87, 6.45 | 2008 - 2010 | Temperate | Germany |
| | DK-Fou | 56.48, 9.59 | 2005 - 2005 | Temperate | Denmark |
| | IE-Ca1 | 52.86, -6.92 | 2004 - 2006 | Temperate | Ireland |
| | IT-BCi | 40.52, 14.96 | 2005 - 2010 | SubTropical | Italy |
| | IT-CA2 | 42.38, 12.03 | 2012 - 2013 | SubTropical | Italy |

Note: The data shown are sourced from FLUXNET2015 (FLUXNET 2015 dataset for micrometeorological measurements), LaThuile (FLUXNET LaThuile 2007 synthesis dataset), and OzFlux (Australian and New Zealand flux research and monitoring networks) (Pastorello et al., 2020). Site names follow the FLUXNET/ICOS convention, using two-letter country codes and three-letter site abbreviations (e.g., CN-Din = Dinghushan, China). The climate classifications are based on the Köppen climate classification system: humid subtropical, Cfa/Cwa; tropical rainforest, Af; subtropical, Csa/Csb; temperate, Cfb/Cfa; and arctic, ET. Coordinates are presented as (longitudes and latitudes) in decimal degrees.

We performed one-factor-at-a-time sensitivity analysis to identify model parameters with the most significant effects on heat fluxes. We tested all the PFT parameters included in the LSM ILS based on previously reported parameter ranges (Poulter et al., 2011, 2015). Each parameter was sampled at four uniformly spaced levels across its plausible range, and with all other



parameters held constant, each parameter was individually perturbed to drive 1-year ILS simulations of sensible and latent heat fluxes. Further, we computed the RMSE between default and perturbed flux simulations to rank parameter sensitivities, as shown in **Fig. 2**. Via this analysis, we identified the 12 parameters with the most significant impact on the outputs selected for calibration.


**Table 2. Key plant functional type parameters and their definitions, default values, and ranges.**

| Parameters | Explanation | Default Value (Evergreen Forest) | Default Value (Woodland) | Default Value (Grassland) | Default Value (Cultivation) | Unit | Range |
|---|---|---|---|---|---|---|---|
| vegh | Vegetation height | 35 | 20 | 1 | 1 | m | [0.5, 40] |
| rlfv | Leaf albedo (visible) | 0.1 | 0.07 | 0.11 | 0.11 | - | [0.05, 0.3] |
| rlfn | Leaf albedo(near-infrared) | 0.45 | 0.4 | 0.58 | 0.58 | - | [0.3, 0.7] |
| tlfn | Leaf trans. (near-infrared) | 0.25 | 0.15 | 0.25 | 0.25 | - | [0.1, 0.5] |
| vgcov | Vegetation coverage | 1 | 1 | 1 | 1 | - | [0.1, 1] |
| cdl | Leaf Exchange coefficient (vapor) | 0.11 | 0.111 | 9.82d-2 | 0.098 | s m-2 | [0.01, 0.5] |
| chl | Leaf Exchange coefficient (thermal) | 0.0274 | 0.0277 | 2.46d-2 | 0.0246 | s m-2 | [0.01, 0.1] |
| vmax0 | Maximum Rubisco capacity | 6.00E-05 | 6.00E-05 | 6.00E-05 | 6.00E-05 | mol m-2 s-1 | [1e-6, 1e-4] |
| gradm | Stomatal conductance slope | 9 | 7.5 | 4 | 9 | - | [1, 20] |
| binter | Minimum stomatal conductance | 0.01 | 0.01 | 0.04 | 0.01 | mol m-2 s-1 | [0.001, 0.1] |
| effcon | Intrinsic quantum efficiency | 0.08 | 0.08 | 0.05 | 0.08 | Mol mol-1 | [0.01, 0.2] |
| psicr | Critical water potential | -200 | -200 | -200 | -200 | kPa | [-500, -50] |



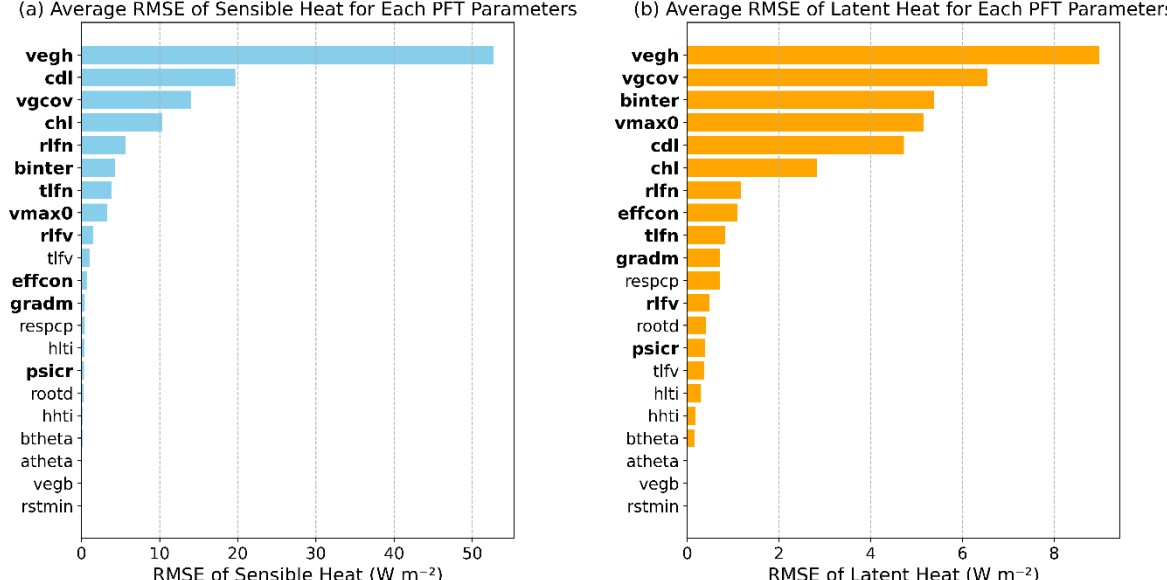

**Figure 2. Comparison of average RMSEs between default and perturbed flux simulations for sensible and latent heat fluxes across**
**PFT parameters(PLUMBER2, ES-LgS Site). The parameters in bold indicate calibration targets.**

A pre-training dataset was generated using the same perturbation strategy that was applied in the sensitivity analysis to enable the surrogate model better capture parameter information. We perturbed the 12 calibration parameters listed in **Table 2** and concatenated the simulation results from this perturbed dataset with those generated using the default parameter set.
MdPL training was performed using the PyTorch framework on an NVIDIA A100 GPU. The LSM simulations were supported by an Intel Xeon Gold 6126 CPU utilizing six nodes for parallel processing.

### 2.4.2 ILS

The land surface model used in this study was an ILS specifically designed to simulate the interactions between the terrestrial surface and the atmosphere (Guo et al., 2021; Nitta et al., 2020). This ILS offered the possibility to evaluate the
performance of the calibrated parameters and was used to conduct sensitivity analysis experiments.

### 2.4.3 Experiments

To comprehensively validate the performance and interpretability of the proposed multitask surrogate model in terms of accuracy and adaptability, we designed two experiments focusing on parameter calibration. Further, to evaluate its transferability, we applied LOOCV, with each site excluded from validation while training the remaining sites (Lumumba et
al., 2024).



Experiment 1 was performed to verify the performance of the multiple-task surrogate model with respect to parameter calibration. We this experiment involved 20 PLUMBER2 sites spanning four PFTs: evergreen forest, woodland, grassland, and cultivation. Further, a subset of four evergreen forest sites was used to compare the multitask surrogate with a single-task dPL surrogate. The models were trained within the dPL framework but with different architectures.

Multiple-task surrogate model:

- Shared layer: 4-layer LSTM with a hidden size of 128.
- Task-specific layer: 2-layer LSTM with a hidden size of 128.
- Time sequence length: 48-time steps (representing 2 days of data).
- Total trainable parameters: 1,096,962.

Single-task surrogate model

- 6-layer LSTM with a hidden size of 190.
- Time sequence length: 48-time steps.
- Total trainable parameters: 980,400.

Parameter generator, $g_z$

• 4-layer LSTM with a hidden size of 128.

- Time sequence length: 48-time steps.

As described in Sect. 2.4.1, we combined default and perturbed-parameter ILS outputs to obtain a surrogate training dataset. Both surrogate models were trained for 200 epochs using the Adam optimizer, with the learning rate (lr) set at 0.005. For the multiple-task surrogate model, a rolling training strategy in which the learning rate for the shared layer was set to one-third
of the base learning rate (i.e., 0.005/3), was employed.

To train $g_z$, forcing, flux observations, and site attributes were used as inputs, employing the same optimizer settings (Adam optimizer lr = 0.005 and 2000 epochs) as were employed for the surrogate models. The calibrated parameters were then fed into the ILS to simulate sensible and latent heat fluxes, which thereafter, were compared to observational data using RMSE and Pearson's $R$ to quantify calibration gains. This experiment highlighted the ability of the MdPL to enhance calibration
across PFTs by leveraging shared representations without sacrificing task-specific accuracy.

Experiment 2 was performed to benchmark the MdPL-calibrated ILS against standard LSMs and data-driven approaches. The goals of this experiment was to: (1) test whether a deep-learning-calibrated physical model can exceed pure ML accuracy and (2) evaluate MdPL-calibrated ILS against established LSMs. Thus, we used the PLUMBER2 outputs from 16 sites to perform this comparison. Four sites (AR-SLu, CN-Din, PT-Mi1, and JP-SMF) without LSTM benchmarks were
omitted. Thus, we evaluated:

1. ILS_MdPL (MdPL-calibrated ILS)
2. ILS_ORI (default ILS)
3. CLM5 (NCAR, 2020)



4. JULES (McNeall et al., 2024)
5. GFDL (Shevliakova et al., 2024)
6. Noah (Ek et al., 2003)
7. LSTM (Hochreiter & Schmidhuber, 1997)

For SFE, a total of 10 sites (DE-Meh, DK-Lva, IE-Dri, DK-Fou, DE-Seh, IE-Ca1, AU-Emr, PL-wet, IT-BCi, and IT-CA2) were used for comparison given that the method requires observed ground heat flux data to calculate latent and sensible heat
fluxes, which are not available for all sites in the PLUMBER2 dataset. The calibrated and default versions of the ILS and other models were evaluated using the same inputs and outputs (sensible and latent heat fluxes) for direct comparison. Additionally, by comparing ILS_dPL with other LSMs and LSTM, we demonstrated the potential of the calibrated LSM to outperform purely deep-learning approaches.

Additionally, in experiment 3 we evaluated the transferability of MdPL via leave-one-out cross-validation within each of the
three PFT categories (evergreen forest, woodland and cropland). In each iteration, one site was withheld and the calibrated parameter vectors from all other sites of the same PFT were averaged to form an ensemble mean. To avoid bias from parameters exhibiting large inter-site variability (namely vegh and vgcov), these two were held at their default values rather than the ensemble mean. Finally, this hybrid parameter set was used to drive ILS at the held-out site, and the resulting performance quantified the extent to which an ensemble-derived mean can be transferred to an ungauged location within that
PFT category.

## 3 Results and discussion

### 3.1 Site-specific Parameter Calibration

The calibrated parameters were fed into the ILS to simulate sensible and latent heat fluxes, and the simulation results before and after calibration were compared with observation data. **Fig. 3** shows Taylor diagrams for the normalized RMSEs, R
values, and normalized STDs for all sites against observation data. The parameter calibration results for each site are presented in **Table A1-A4.**









**Figure 3. Taylor diagrams showing normalized RMSEs, R, and normalized STDs for sensible heat and latent heat outputs across all sites. (a) Sensible heat simulations for four PFT sites. (b) Latent heat simulations. OBS, observations of sensible and latent heat; ILS_ORI, outputs from ILS with default PFT parameters; ILS_MdPL, outputs from ILS with PFT parameters corrected using MdPL; Δ, percentage change.**

Considering default PFT parameters across all sites, **Fig. 3(a)** shows that MdPL calibration significantly improved model performance. The most significant decrease in RMSE was observed for evergreen forest sites, with the average reduction rate at 24%, whereas a minor reduction rate was observed for grassland sites, with the RMSE only decreasing slightly at 7.72%. Further, cultivation sites exhibited the highest improvement in R (3.02%), whereas woodland showed the lowest increase (1.2%). Furthermore, the normalized STDs for sensible heat outputs at most sites were closer to the observed values, demonstrating that the MdPL significantly enhanced the performance of the complex model in sensible heat simulations.

**Figure 3(b)** shows latent heat simulation results. From this figure, it is evident that RMSEs decreased across all sites, with evergreen forest sites showing the most significant reduction (20.01%), whereas woodland sites only showed minor reduction (8.74%). Further, while other PFTs showed improvements in R values (e.g., 7.05%, 2.66%, and 1.2% for evergreen forest, woodland, and grassland sites, respectively), the R value for cultivation sites decreased significantly (-4.77%). This observation could be explained as follows:

1. Model parameters for R were not optimized given that the model did not explicitly include the loss function during calibration.

2. Human activities and seasonal changes strongly influence PFT parameters at cultivation sites. Given that the ILS model uses fixed PFT parameters rather than dynamic parameters over time, it failed to accurately capture the temporal variability of latent heat fluxes.

Nevertheless, the overall trend demonstrated that the MdPL framework consistently showed improved performance across most sites and output variables, confirming its efficacy in parameter calibration in LSMs. Additionally, relative to previous studies that primarily focused on a single watershed or specific vegetation types, the MdPL framework employed in this study covered four major PFTs and 20 sites, demonstrating its robustness across diverse scenarios.

### 3.2 Comparison of Multiple Task and Single Task Surrogate Models

To validate the superior performance of the Multiple Task surrogate model (ILS_MdPL) relative to the Single Task surrogate model (ILS_dPL), we selected four evergreen forest sites and performed parameter calibration using both approaches. The evaluation metrics obtained (**Table 3**) indicated that both models outperformed the default parameter set in terms of RMSE and R. For sensible heat, the average RMSE obtained for ILS_MdPL was slightly higher than that obtained for the ILS_dPL. However, ILS_MdPL showed a lower RMSE for latent heat than ILS_dPL. For R, ILS_MdPL outperformed ILS_dPL in terms of both sensible and latent heat fluxes. These observations indicated that even after sharing





parameters across multiple tasks, ILS_MdPL still maintained strong collaborative effects between key variables, excelling in
latent heat flux simulations. The parameter calibration results for each site are presented in **Table S5**.

**Table 3. Evaluation metrics (RMSE and R) for parameter calibration performance based on the ILS_MdPL and ILS_dPL frameworks at four evergreen forest sites.**

| Site Name | ILS_ORI | | | | ILS_MdPL | | | | ILS_dPL | | | |
|---|---|---|---|---|---|---|---|---|---|---|---|---|
| | RMSE | | R | | RMSE | | R | | RMSE | | R | |
| | Sensible heat (W m$^{-2}$) | Latent heat (W m$^{-2}$) | Sensible heat | Latent heat | Sensible heat (W m$^{-2}$) | Latent heat (W m$^{-2}$) | Sensible heat | Latent heat | Sensible heat (W m$^{-2}$) | Latent heat (W m$^{-2}$) | Sensible heat | Latent heat |
| CN-Din | 47.49 | 58.06 | 0.84 | 0.78 | 37.26 | 48.3 | 0.84 | 0.81 | 37.21 | 54.73 | 0.82 | 0.74 |
| ID-Pag | 52.38 | 60.26 | 0.85 | 0.87 | 24.37 | 39.78 | 0.92 | 0.96 | 23.74 | 40.46 | 0.91 | 0.95 |
| PT-Esp | 70.09 | 43.62 | 0.78 | 0.77 | 66.65 | 39.11 | 0.82 | 0.80 | 63.15 | 40.24 | 0.83 | 0.78 |
| PT-Mi1 | 55.43 | 27.12 | 0.92 | 0.63 | 43.01 | 24.03 | 0.92 | 0.71 | 41.61 | 22.57 | 0.92 | 0.62 |
| **Mean** | **56.35** | **47.26** | **0.85** | **0.76** | **42.82** | **37.8** | **0.88** | **0.82** | **41.43** | **39.5** | **0.87** | **0.77** |

Note: The comparison of model performance with respect to the simulation of sensible heat and latent heat fluxes was relative to the
default parameter set (ILS_ORI).





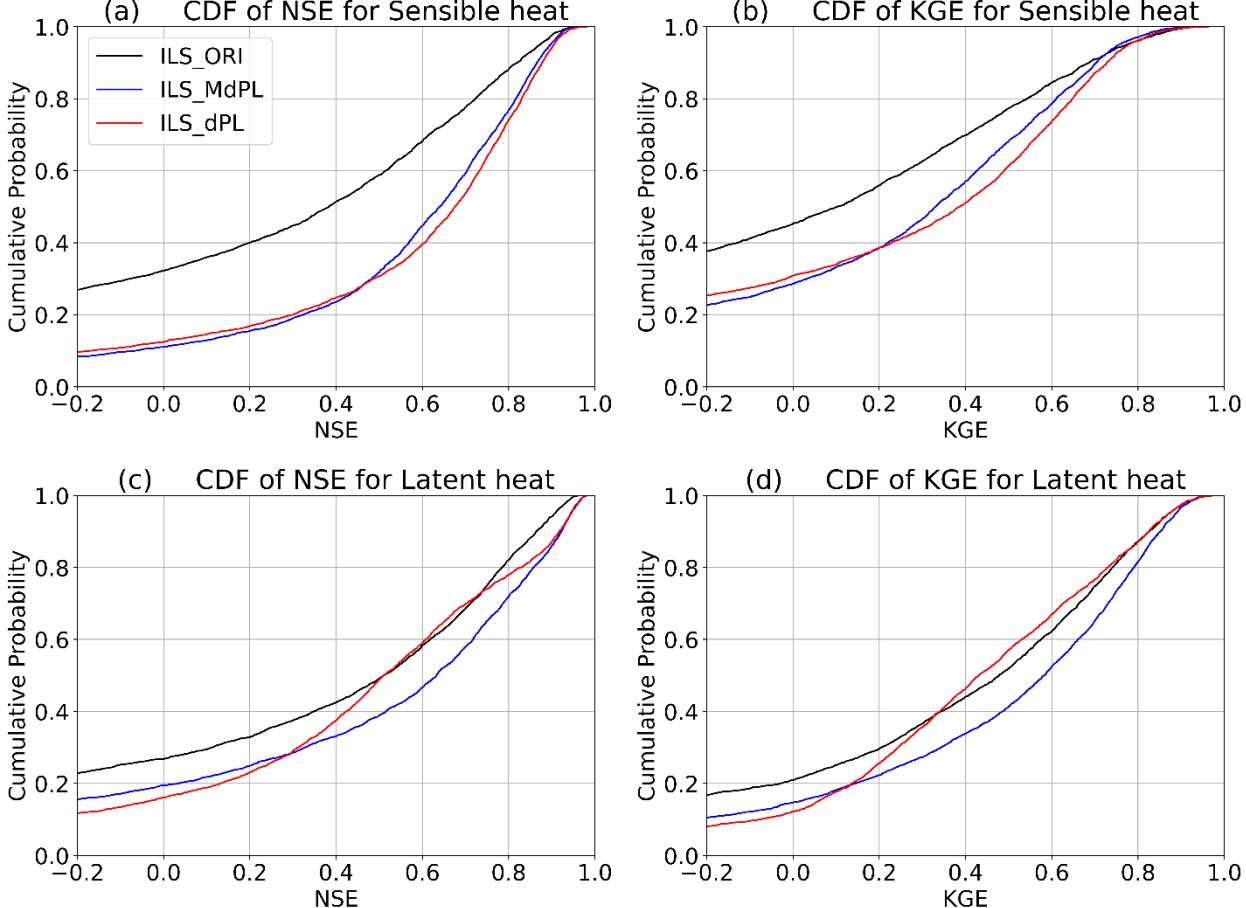

**Figure 4. Cumulative distribution functions (CDFs) of daily NSE and KGE for sensible and latent heat outputs across all sites. (a)**
**NSE- and (b) KGE-based sensible heat flux performance comparison between multi-task (ILS_MdPL) and single-task (ILS_dPL)**
**surrogate models. (c) NSE- and (d) KGE-based latent heat flux performance comparison between multi-task (ILS_MdPL) and**
**single-task (ILS_dPL) surrogate models. The curves on the right in both panels represent better performing models.**

To assess the performance of the models with respect to hydrological metrics, we plotted a graph to show the cumulative
distribution function (CDF) of daily NSE and KGE, as shown in **Fig. 4**. Regarding sensible heat, the performances of
ILS_MdPL and ILS_dPL were relatively similar. Specifically, in the low NSE range (-0.2 to 0.45) and low KGE range (-0.2
to 0.2), ILS_MdPL outperformed ILS_dPL, indicating superior robustness for ILS_MdPL in lower-accuracy simulations.
For latent heat, the performance of ILS_dPL was lower than that of the default parameter set, ILS_ORI. However, the CDF
curves of ILS_MdPL shifted further to the right, implying that it exhibited good scalability and could adapt to multiple
variable requirements. Thus, it showed an enhanced ability to comprehensively fit complex systems.



In summary, the multiple-task surrogate model was advantageous in terms of parameter calibration, particularly in multi-output scenarios, showing higher stability and flexibility and demonstrating a high potential for application in parameter calibration in complex LSMs.



**3.3 Comparative Evaluation of Calibrated ILS and Widely Used LSMs Against LSTM**

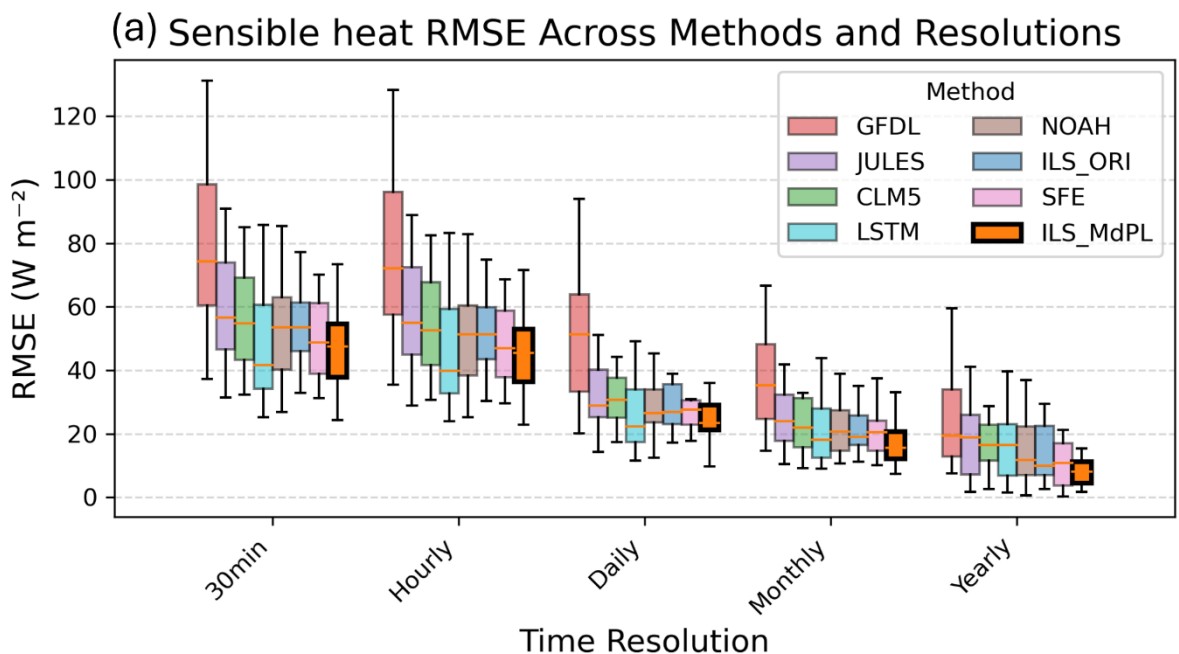

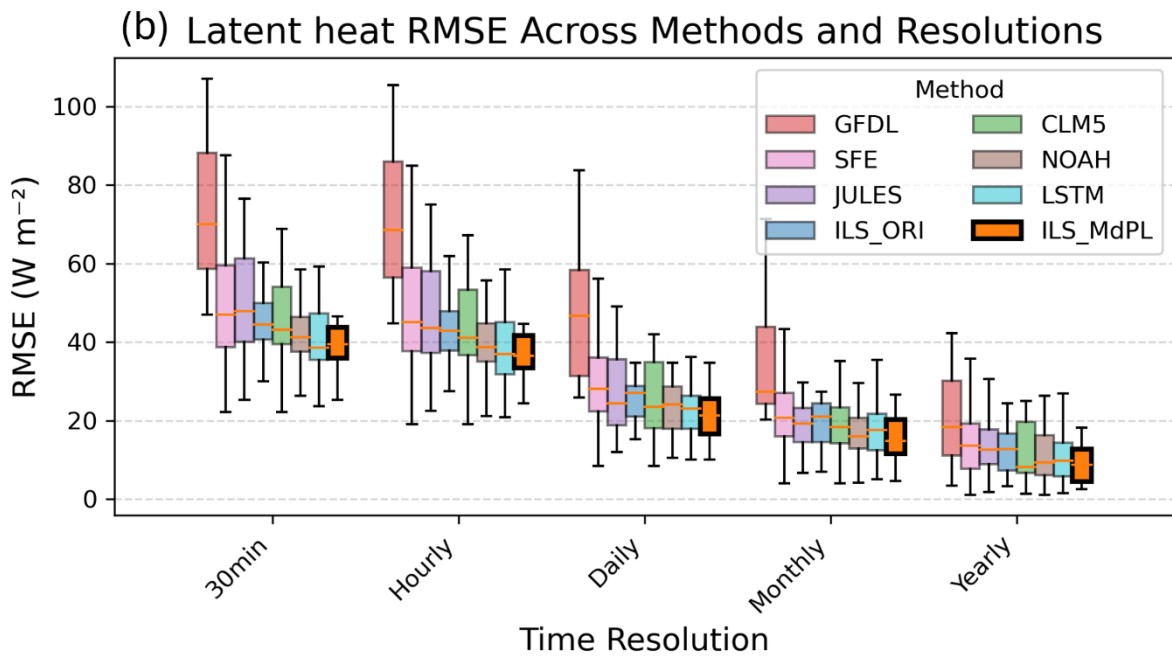


**Figure 5. Comparison of RMSEs for sensible and latent heat fluxes across methods and resolutions. (a) Box plots of RMSE distributions for sensible heat flux. The number of LSM and SFE sites were 16 and 10, respectively. (b) Box plots showing the**




**distribution of RMSEs for latent heat flux. The number of LSM and SFE sites were 16 and 10, respectively. The yellow line represents the median. The upper and lower box boundaries indicate 75% and 25% interquartile ranges, respectively.**

**Figure 5** shows the distribution of RMSEs across different time resolutions. In this study, we primary focused on comparing the calibrated land surface model (ILS_MdPL) with the deep learning-based surrogate model (LSTM), as this comparison provided more insightful implications than those obtained by merely comparing calibrated and uncalibrated LSMs.

At the finest time resolution (30 min), ILS_MdPL and LSTM exhibited nearly identical RMSEs, but as the temporal resolution became coarser (from hourly to yearly time intervals), the advantages of the ILS_MdPL became more

pronounced. Notably, it consistently outperformed the LSTM, showing lower RMSEs for both sensible and latent heat fluxes, highlighting the robustness of this physically constrained calibration approach, particularly at larger temporal scales. The difference in latent heat flux was even more substantial. At temporal resolution of 30 min, ILS_MdPL achieves a lower RMSE (41.45 W m-2) than LSTM (44.02 W m-2), and this difference widened as the time resolution increased, emphasizing the strength of ILS_MdPL in capturing the complex variability of latent heat via physically meaningful calibration.

Regarding sensible heat, the accuracy of SFE was similar to that of ILS_MdPL. However, for latent, it performed poorly at high-frequency resolutions (e.g., 30 min), but shows considerable improvement as the resolution increased. Specifically, at the monthly scale, its performance in predicting sensible heat ranked second only to that of ILS_MdPL, implying that it outperformed all the other traditional LSMs. This observation suggested that even purely empirical models, such as SFE, can be competitive at aggregated scales, possibly owing to their ability to exploit large-scale patterns without physical

complexity.

In summary, all the evaluation metrics and time resolutions consistently demonstrated the effectiveness and robustness of the proposed MdPL calibration method. Relative to the default parameter set and several widely used LSMs, the ILS_MdPL framework showed a significantly improved model accuracy, particularly in reproducing both sensible and latent heat fluxes. While LSTM showed competitive performance at fine temporal resolutions, its advantages diminished at coarser scales, with

the physically constrained ILS_MdPL showing superior performance by a notable margin. Even though empirical methods, such as the SFE, show limited performance in capturing high-frequency dynamics, their performance at aggregated time scales suggested that they have potential for use in simplified large-scale assessments. Overall, the proposed approach balanced physical interpretability, multi-output calibration capability, and computational efficiency, offering a promising direction for advancing land surface modeling and parameter calibration frameworks.

**3.4 Transferability of MdPL Parameter Calibrations**

To assess the transferability and generalizability of MdPL-derived parameter calibrations, we separately performed leave-one-out cross-validation (ILS_MdPL_LOOCV) for three PFT classes: evergreen forest (n = 4; PT-Mi1, CN-Din, ID-Pag, PT-Esp), Woodland (n = 5; AR-SLu, DE-Meh, US-Bar, JP-SMF, UK-PL3), and Cultivation land (n = 3; DK-Fou, IE-Ca1, DE-Seh), and reported all monthly KGE results as mean ± STD over the corresponding n experiments. Thereafter, each

LOOCV-derived parameter set was used to drive ILS simulations, and the resulting mean ± STD values of monthly KGE for





sensible heat and latent heat fluxes were compared against the KGE values obtained for both the original ILS parameterization (ILS_ORI) and site-specific MdPL parameterization (ILS_MdPL), as summarized in **Table 4**.


**Table 4. Mean ± STD of KGE values for sensible heat and latent heat fluxes simulated using the: (1) original ILS parameter set (ILS_ORI), (2) leave-one-out–calibrated MdPL (ILS_MdPL_LOOCV), and (3) site-specific MdPL (ILS_MdPL), averaged over LOOCV experiments (n = 4 for Evergreen forest; n = 5 for Woodland; n = 3 for Cultivation sites).**

| | Sensible heat | | | Latent heat | | |
|---|---|---|---|---|---|---|
| | ILS_ORI | ILS_MdPL_LOOCV | ILS_MdPL | ILS_ORI | ILS_MdPL_LOOCV | ILS_MdPL |
| Evergreen Forest (n = 4) | -0.05±0.98 | 0.32±0.40 | **0.42**±0.47 | 0.30±0.40 | 0.63±0.17 | **0.73**±0.05 |
| Woodland (n = 5) | 0.29±0.24 | 0.32±0.18 | **0.45**±0.38 | **0.65**±0.21 | 0.54±0.20 | 0.63±0.18 |
| Cultivation (n = 3) | -0.29±0.66 | -0.40±0.39 | **0.49**±0.13 | -0.38±1.03 | -0.65±1.28 | **0.42**±0.42 |

For evergreen forest sites, LOOCV-derived parameters generated higher monthly KGE values than ILS_ORI for both sensible (0.32 vs. -0.05) and latent (0.63 vs. 0.3) heat fluxes, even though both remained below the ILS_MdPL-based values.

Woodland sites exhibited comparable KGE values between ILS_MdPL_LOOCV and ILS_ORI for both fluxes (0.32 vs. 0.29 for sensible heat; 0.54 vs. 0.65 for latent heat). In contrast, cultivation sites showed a negative mean KGE under both ILS_ORI and LOOCV, whereas ILS_MdPL achieved values higher than 0.4, demonstrating poor transferability for this PFT. These findings indicated that MdPL-derived parameters transfer effectively among relatively stable and natural PFTs (Evergreen forest and Woodland), but are hindered in heavily managed systems, such as cultivation, characterized by greater

environmental and management heterogeneity (Martin & Isaac, 2018).



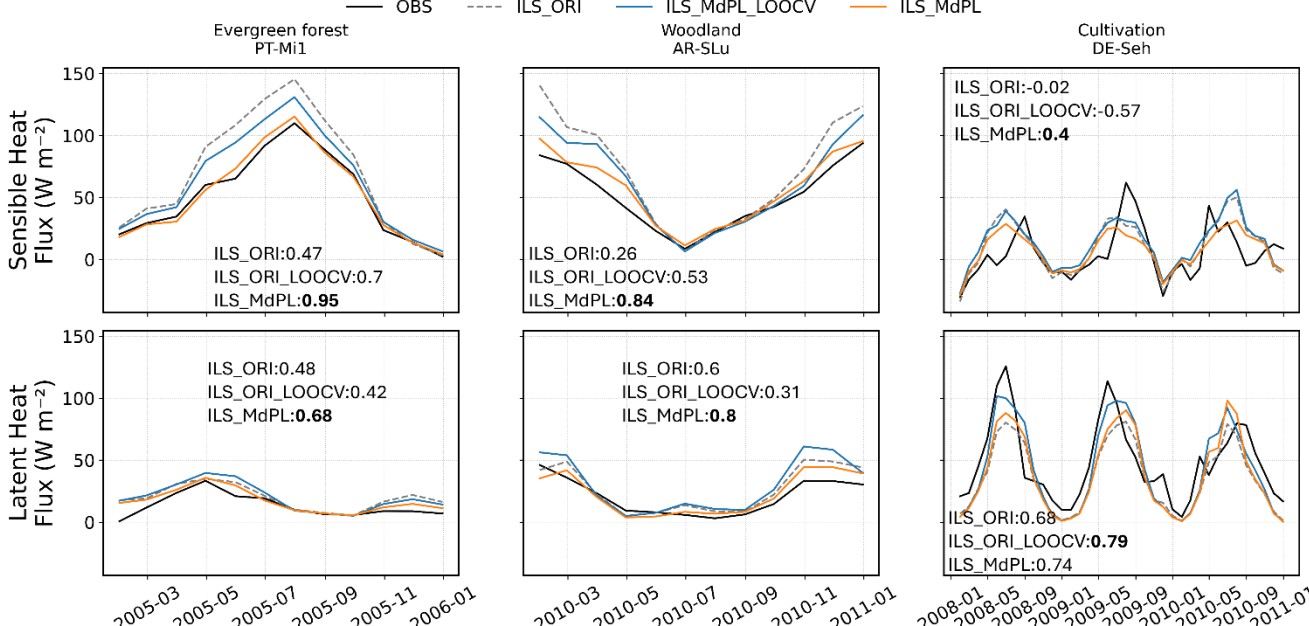

**Figure 6. Comparison of monthly-scale KGE values for ILS_ORI, ILS_MdPL_LOOCV, and ILS_MdPL across Evergreen Forest site PT-Mi1, Woodland site AR-SLu, and Cultivation Site DE-Seh.**

**Figure 6** shows monthly KGE curves for ILS_ORI, ILS_MdPL_LOOCV, and ILS_MdPL across the three PFT sites
considered. For evergreen forests and woodlands, the performance ranking consistently followed the trend: ILS_ORI <
LOOCV < MdPL, with LOOCV showing a narrow gap between observations, and MdPL providing the closest fit.
Conversely, for cultivation sites, ILS_ORI and LOOCV curves overlapped in the negative-KGE regime, while the sensible
heat simulation performance of MdPL increased to approximately 0.4, underscoring the difficulty of parameter transfer in
agricultural systems. This systematic decline in transferability from forests to cultivation land could be attributed to land
management heterogeneity, phenological variability, and environmental heterogeneity, which limit model generalization
(Hoppe et al., 2024). Overall, these findings underscore the fact that the benefits of data-driven calibration are contingent on
the intrinsic transferability of the site characteristics embedded within the training data.

Despite these promising results, this study had some limitations. First, the current calibration primarily targeted energy
fluxes (sensible and latent heat), whereas other critical LSM outputs, such as surface temperature and soil moisture content,
were not evaluated. Additionally, the ILS model employed fixed PFT parameters, which may not adequately reflect temporal
variability in vegetation, particularly for cultivation sites influenced by human management activities. The increased
complexity of the multitask surrogate model, even though beneficial in terms of performance, may also pose challenges in
terms of computational cost and overfitting in data-scarce regions. Finally, the systematic decline in transferability from
forests to cultivation sites highlighted a key limitation of the MdPL: the benefits of data-driven calibration depend critically
on the intrinsic similarity of the site characteristics represented in the training data.



## 4. Conclusion

In this study, we introduced MdPL, a deep-learning-based calibration framework that couples a multitask neural surrogate with a differentiable parameter generator to improve LSM performance. The application to this framework to 20 PLUMBER2 sites spanning four PFTs (evergreen forest, woodland, grassland, cultivation) achieved substantial gains: a 15%
reduction in RMSE for sensible and latent heat fluxes relative to the default model, and accuracy similar to or superior to those of leading LSMs (CLM5, JULES, Noah, GFDL), and an LSTM benchmark. Via one-factor-at-a-time sensitivity analysis, the 12 most influential parameters were identified, and LOOCV experiments showed robust transferability for forests and woodlands. However, transferability performance for cultivation sites was limited by fixed PFT representations. Regardless of this limitation, the proposed model offers a scalable and efficient approach for enhancing parameter
calibration in LSMs. In future, it will be necessary to focus on integrating dynamic vegetation processes, expand calibrated outputs to include hydrological and thermal variables, and evaluate the model generalizability under extreme climatic events and broader spatial domains. Additionally, reducing computational cost while maintaining interpretability and physical consistency will be key to operationalizing this framework in large-scale Earth system modeling efforts.

**Appendix A**

Table A1. MdPL parameter calibration results for Evergreen Forest. Numbers shown in bold font exceed the specified lower or upper limits and have been set to the corresponding bound.

| Evergreen Forest | default | CN-Din | ID-Pag | PT-Esp | PT-Mi1 |
|---|---|---|---|---|---|
| vegh | 35 | 20.22 | 16.30 | 6.60 | **0.5** |
| rlfv | 0.1 | 0.03 | 0.05 | 0.05 | 0.16 |
| rlfn | 0.45 | 0.42 | 0.36 | 0.34 | 0.63 |
| tlfn | 0.25 | 0.33 | 0.19 | 0.14 | 0.25 |
| vgcov | 1 | 0.96 | 0.79 | 0.11 | 0.84 |
| cdl | 0.11 | 0.37 | 0.36 | 0.29 | 0.41 |
| chl | 0.0274 | **0.01** | **0.01** | **0.01** | 0.0131 |
| vamx0 | 6.00E-05 | 1.23E-05 | -9.35E-06 | 4.36E-05 | 2.15E-06 |
| gradm | 9 | 10.93 | 1.40 | **1** | 3.98 |
| binter | 0.01 | **0.001** | 0.07 | 0.02 | 0.01 |
| effcon | 0.08 | **0.01** | **0.01** | 0.03 | 0.06 |
| psicr | -200 | -272.41 | -541.77 | -313.03 | -115.21 |





Table A2. MdPL parameter calibration results for Cultivation. Numbers shown in bold font exceed the specified lower or upper limits and have been set to the corresponding bound.

| Cultivation | default | DK-Fou | DE-Seh | IE-Ca1 | IT-CA2 | DK-Ris | IT-Bci |
|---|---|---|---|---|---|---|---|
| vegh | 1 | **0.5** | 2.06 | 2.19 | **0.5** | 1.66 | 1.13 |
| rlfv | 0.11 | 0.11 | 0.14 | 0.07 | 0.18 | 0.04 | 0.04 |
| rlfn | 0.58 | 0.23 | 0.45 | 0.22 | 0.33 | 0.32 | 0.30 |
| tlfn | 0.25 | 0.19 | 0.30 | **0.1** | 0.23 | **0.1** | 0.13 |
| vgcov | 1 | 0.49 | 0.78 | 0.86 | 0.86 | 0.39 | 0.82 |
| cdl | 0.098 | 0.212 | 0.310 | 0.442 | 0.252 | 0.261 | 0.110 |
| chl | 0.0246 | 0.0324 | 0.0152 | **0.01** | 0.0268 | 0.0124 | 0.0432 |
| vamx0 | 6.00E-05 | 2.73E-05 | 6.57E-05 | 1.00E-6 | 5.41E-05 | 3.72E-05 | 5.04E-05 |
| gradm | 9 | 4.69 | 13.30 | 9.54 | 10.61 | 7.10 | 11.01 |
| binter | 0.01 | **0.001** | 0.07 | **0.001** | 0.05 | 0.01 | 0.05 |
| effcon | 0.08 | 0.06 | 0.01 | 0.04 | 0.06 | 0.11 | 0.03 |
| psicr | -200 | -370.56 | -237.23 | -367.18 | -171.21 | -317.93 | -215.11 |




Table A3. MdPL parameter calibration results for Grassland. Numbers shown in bold font exceed the specified lower or upper limits and have been set to the corresponding bound.

| Grassland | default | AU-Emr | DK-Lva | PL-wet | CN-Dan | CN-Du2 | IE-Dri |
|---|---|---|---|---|---|---|---|
| vegh | 1 | **0.5** | 6.59 | 3.01 | 0.63 | **0.5** | 0.61 |
| rlfv | 0.11 | **0.05** | 0.17 | **0.05** | **0.05** | **0.05** | 0.12 |
| rlfn | 0.58 | **0.3** | 0.41 | **0.3** | 0.43 | 0.33 | 0.43 |
| tlfn | 0.25 | **0.1** | 0.27 | **0.1** | 0.15 | **0.1** | 0.20 |
| vgcov | 1 | 0.40 | 0.59 | 0.53 | 0.57 | 0.57 | 0.71 |
| cdl | 9.82d-2 | **0.01** | 0.270 | 0.081 | 0.242 | 0.146 | 0.207 |
| chl | 2.46d-2 | **0.01** | 0.0600 | **0.01** | 0.0106 | 0.0327 | **0.01** |
| vamx0 | 6.00E-05 | **1.00E-6** | **1.00E-6** | 2.81E-05 | 4.85E-05 | 4.51E-05 | 2.22E-05 |
| gradm | 4 | 4.67 | 1.05 | 5.51 | 9.84 | 8.62 | 5.05 |
| binter | 0.04 | **0.001** | 0.02 | 0.04 | 0.04 | 0.04 | 0.06 |
| effcon | 0.05 | 0.02 | 0.12 | 0.08 | 0.07 | 0.04 | 0.07 |
| psicr | -200 | -490.19 | -258.58 | -484.86 | -243.03 | -327.12 | -269.51 |



Table A4. MdPL parameter calibration results for Woodland. Numbers shown in bold font exceed the specified lower or upper limits and have been set to the corresponding bound.

| Woodland | default | AR-SLu | JP-SMF | UK-PL3 | CN-Cha | DE-Meh | US-Bar |
|---|---|---|---|---|---|---|---|
| vegh | 20 | 5.60 | 27.23 | **0.5** | 1.28 | 4.44 | 27.86 |
| rlfv | 0.07 | **0.05** | **0.05** | **0.05** | **0.05** | 0.12 | **0.05** |
| rlfn | 0.4 | **0.3** | 0.40 | 0.36 | 0.42 | 0.42 | 0.42 |
| tlfn | 0.15 | 0.25 | **0.1** | **0.1** | 0.19 | **0.1** | 0.12 |
| vgcov | 1 | 0.63 | 0.71 | 0.60 | 0.99 | 1.00 | 0.90 |
| cdl | 0.111 | 0.134 | 0.387 | 0.206 | 0.108 | 0.275 | 0.337 |
| chl | 0.0277 | **0.01** | **0.01** | 0.0110 | 0.0216 | 0.0291 | 0.0320 |
| vamx0 | 6.00E-05 | **1.00E-6** | 6.25E-05 | 2.47E-05 | 5.83E-05 | 3.17E-05 | 7.73E-06 |
| gradm | 7.5 | 7.65 | 3.99 | 4.80 | 10.35 | 3.63 | **1** |
| binter | 0.01 | **0.001** | 0.05 | 0.02 | 0.03 | 0.02 | 0.01 |
| effcon | 0.08 | 0.14 | 0.03 | 0.04 | 0.04 | 0.07 | **0.01** |
| psicr | -200 | -123.19 | -187.25 | -581.03 | -244.42 | -496.00 | -389.32 |

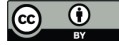



Table A5. dPL parameter calibration results for Evergreen Forest. Numbers shown in bold font exceed the specified lower or upper limits
and have been set to the corresponding bound.

| Evergreen Forest From dPL | default | CN-Din | ID-Pag | PT-Esp | PT-Mi1 |
|---|---|---|---|---|---|
| vegh | 35 | 24.26 | 3.21 | 20.87 | 3.88 |
| rlfv | 0.1 | 0.08 | 0.15 | 0.02 | 0.14 |
| rlfn | 0.45 | 0.41 | 0.41 | 0.34 | 0.41 |
| tlfn | 0.25 | 0.26 | 0.19 | 0.21 | 0.27 |
| vgcov | 1 | 0.61 | 0.82 | 0.22 | 0.58 |
| cdl | 0.11 | 0.39 | 0.21 | 0.35 | 0.44 |
| chl | 0.0274 | **0.01** | **0.01** | 0.0155 | 0.0123 |
| vamx0 | 6.00E-05 | 5.55E-05 | 1.55E-05 | 6.51E-05 | -8.14E-07 |
| gradm | 9 | 3.82 | **1** | 6.09 | 0.66 |
| binter | 0.01 | 0.01 | 0.05 | 0.03 | 0.00 |
| effcon | 0.08 | **0.01** | **0.01** | 0.07 | 0.08 |
| psicr | -200 | -294.71 | -556.66 | -276.42 | -410.23 |

**Data Availability**

The PLUMBER2 benchmarking dataset used for model evaluation is publicly available at the Australian Research Data
Commons: https://researchdata.edu.au/plumber2-forcing-evaluation-surface-models/1656048. The underlying
FLUXNET2015, LaThuile and OzFlux tower data are provided at https://fluxnet.org/. All data generated and analyzed
during this study are publicly available in the Zenodo repository at https://doi.org/10.5281/zenodo.15753067. Detailed
instruction are provided in the repository README. These resources are released under a CC-BY 4.0 license.

**Code Availability**

All custom code for data preprocessing, model training, ILS simulation, calibration experiments, and figure generation is
publicly available at Zenodo: https://doi.org/10.5281/zenodo.15748737. Detailed instructions, environment specification,
and scripts are provided in the repository README.

**Author Contributions**

WX conceptualized the study, developed the MdPL framework, ran the ILS simulations and analyses, and drafted the
manuscript. HL carried out data preprocessing (PLUMBER2), conceptualized the study, and contributed to manuscript



revision. KY supervised the research, provided scientific guidance, and contributed to manuscript revision. All authors discussed the results and approved the final version.

**Financial support**

This study was supported by the water environment and resource research project at the Earth Observation Research Center,
Japan Aerospace Exploration Agency (JAXA EORC). This work was supported by the Japan Society for the Promotion of Science (JSPS) via Grants-in-Aid 22H04938 and 21H05002; the Japan Science and Technology Agency (JST) via the "New Social Challenges" project (Grant JPMJMI24I1).

**Competing Interests**

The authors declare that they have no conflict of interest.

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
