# Peer review of "Enhancing Parameter Calibration in Land Surface Models Using a Multi-Task Surrogate Model within a Differentiable Parameter Learning Framework"

_EGUsphere, 2025_

## Referee Comment (RC1)

**Enhancing Parameter Calibration in Land Surface Models Using a Multi-Task Surrogate Model within a Differentiable Parameter Learning Framework**

By Xie et al.

The paper develops a multi-task differentiable parameter learning (MdPL) framework as a unified calibration approach for land surface models (LSMs). It builds on a previous deep learning scheme, dPL, that directly embeds parameter estimation within model training, and improves its design while also enabling multi-output calibration, with latent heat flux and sensible heat flux as the outputs.

MdPL consists of a multi-task surrogate model of LSM and a parameter generator. It was used in this stuy to calibrate the Integrated Land Simulator (ILS) using site observations representing four functional types. Different experiments were used to assess the performance of the calibrated MdPL-calibrated ILS and compare with the original out-of-sample observation, dPL simulations, ILS simulations, and multiple LSMs.

I support this work as an important step in exploring how AI can improve LSMs. However, I have three major comments that may enhance the reliability and clarity of the performance conclusions.

- First, the study uses only a limited number of sites spanning a few years, which weakens the strength of the conclusions. The PLUMBER2 dataset includes 170 sites, yet only 20 were used here across four PFTs, with each PFT represented by only a handful of sites, and in some cases results are shown for a single site per PFT.
- Second, the model was calibrated and evaluated using latent and sensible heat flux, but the authors do not discuss other fluxes simulated by LSMs that were not included in the calibration, such as NEE. For example, does the MdPL-ILS improve the simulation of NEE? Showing an example of how the model performs on a new flux like NEE would also be interesting.
- Third, while the authors provide estimated parameters for sites across different PFTs in the appendix, there is no interpretation of whether the new values make physical sense more than the original values. Even if a full explanation is not possible, some discussion would help demonstrate that MdPL captures site relationships in a meaningful way.

In the following specific comments for each section:

**Abstract**

- "Exceeded those of LSTM-based approaches": it is actually a single LSTM approach, not multiple.
- Line 25: The statement "close-to-optimal" overestimates the performance, since the KGE scores were far from 1 (the optimal value).

- Line 27: Replace "Despite its" with "Despite the".

**Introduction**

- Line 51: "However, it is associated with several limitations, particularly in the context of LSMs": the statement neither has a reference nor is it explained.

- Line 103: PLUMBER2 should be introduced when first mentioned.

- Line 103: A reference for ILS is needed.

- Line 104: A reference is needed for each of these LSMs, and the acronyms should be expanded on first mention.

- Lines 91–112: The introduction section includes methodological details and results that are better suited to the Methods or Results sections. For example, you mention the role of gate layers (a methodological detail) and how you pretrained the surrogate, and then describe results such as "accurately captured nonlinear and coupled processes". These belong later in the paper. You could reframe these points as **objectives** instead. For instance: state that the aim is to develop a scalable framework, and that you intend to validate it against observational data.

**Method**

- The way the experiments are explained is a bit confusing. It is worth using subheadings for Experiment 1, Experiment 2, and Experiment 3 and revising the text. Also, some helpful details are missing. For each experiment, indicate how many models you will train and how many simulations each model will generate. Later, in the results, state explicitly that section 3.1 is the results of Experiment 1 and section 3.2 is the results of Experiment 2…
- It would be useful to clarify the differentiable aspect of the parameter generator g_z
- In Figure 1 (b), is the arrow connecting *Parameters* and *Day of Year Embedding* correct? Shouldn't the parameters go directly to the tensor concatenation symbol?

- **Line 125**: The subject of Eq.1 (left-hand side) is missing. Is this the loss function of the surrogate model? This needs to be mentioned before displaying the equation.
- **Line 139**: Again, Eq. 2 is missing the first part. Is this the cost function?
- In this part, you describe the multi-task surrogate model in 1b and wrote: *"Further, differences in scale and variation frequency between static and dynamic features…"* Do the Parameters include static ones?
- **Line 193**: Better to omit *natural* and keep only *variability*.
- **Line 202-203**: What is the baseline model? Is it the ILS or the dPL-ILS?
- **Line 216**: Is *averaging* correct here, or do you mean *including*?

**Results:**

- An evaluation of the surrogate model against ILS (as described in Figure A-1) is important to demonstrate that the surrogate closely approximates the ILS model itself.

**In 3.1 Site-specific Parameter Calibration**

- **Figure 3**: Could you explain how $\Delta R^2$ and $\Delta RMSE$ are calculated? For instance, here $\Delta RMSE = 24\%$. This should indicate that IL_MdPL is worse, since RMSE (error) has increased. This is different from $\Delta R^2$, where larger values indicate improvement in skill. Is $R^2$ the proportion of variance explained, or is it the relative difference in standard deviation (the x-axis)? The Taylor diagram shows results for RMSE, correlation, and standard deviation. In the results you discuss *R* and RMSE, but not standard deviation.

- **Line 317**: Explicitly state that these results are for sensible heat flux.

- **Figure 3**: There is a typo in the title of some subplots (*Latnet*).

- **Line 324-353**: The paragraph mentions *R*, but it seems you are referring to the normalised standard deviation. In the first point explaining the reduction in performance related to *R*, could you clarify what you mean?

- Throughout the results, the use of *significantly* is inconsistent. For example, in line 328 you mention *decreased significantly (-4.77%)*, whereas in line 326 you say *only showed minor reduction (8.74%)*, despite |8.74| being larger than |4.77|.

**In 3.2 Comparison of Multiple Task and Single Task Surrogate Models**

- **Figure 4**: This shows that ILS_MdPL is better for sensible heat but worse than ILS_dPL for latent heat. The conclusion that *"it exhibited good scalability and could adapt to multiple variable requirements… high potential for application in parameter calibration in complex LSMs"* is not strongly demonstrated.

- **Line 391**: *"In summary, all the evaluation metrics and time resolutions consistently demonstrated the effectiveness and robustness of the proposed MdPL."* You state *all the evaluation metrics*, but you only show one. To claim robustness, you need to demonstrate this across multiple metrics. A supplementary plot for other metrics (e.g. R, NSE, or NGE) would be needed to support this claim.

**In 3.4 Transferability of MdPL Parameter Calibrations**

- **Tables 3 and 4**: Please add vertical lines to improve readability. For example, in Table 4, you could place a line separating latent heat and sensible heat.

- **Figure 6**: In the caption, it is clearer to state that these are monthly time-series of sensible heat (top) and latent heat (bottom) simulated by the models, with KGE provided. Why are evaluation results shown for a very short record (one year or less) for the first two sites? Why not use sites with longer coverage?

- **Line 424–426**: *"with LOOCV showing a narrow gap between observations"* does not appear correct for Evergreen Forest or Woodland, for either sensible heat flux or latent heat flux.